# Endothelial Function in Dyslipidemia: Roles of LDL-Cholesterol, HDL-Cholesterol and Triglycerides

**DOI:** 10.3390/cells12091293

**Published:** 2023-05-01

**Authors:** Yukihito Higashi

**Affiliations:** 1Department of Regenerative Medicine, Research Institute for Radiation Biology and Medicine, Hiroshima University, Hiroshima 743-8551, Japan; yhigashi@hiroshima-u.ac.jp; Tel.: +81-82-257-5831; 2Division of Regeneration and Medicine, Medical Center for Translational and Clinical Research, Hiroshima University Hospital, Hiroshima 734-8553, Japan

**Keywords:** endothelial function, dyslipidemia, low-density lipoprotein cholesterol, high-density lipoprotein cholesterol, triglycerides

## Abstract

Dyslipidemia is associated with endothelial dysfunction. Endothelial dysfunction is the initial step for atherosclerosis, resulting in cardiovascular complications. It is clinically important to break the process of endothelial dysfunction to cardiovascular complications in patients with dyslipidemia. Lipid-lowering therapy enables the improvement of endothelial function in patients with dyslipidemia. It is likely that the relationships of components of a lipid profile such as low-density lipoprotein cholesterol, high-density lipoprotein cholesterol and triglycerides with endothelial function are not simple. In this review, we focus on the roles of components of a lipid profile in endothelial function.

## 1. Introduction

Epidemiological studies and clinical trials have shown that the relationships of components of a lipid profile with cardiovascular events are not simple [1,2,3,4,5,6]. Reduction in low-density lipoprotein cholesterol (LDL-C) levels by lipid-lowering therapy should be the lower the better for prevention of cardiovascular events in patients with hypercholesterolemia. However, it is likely that lower levels of LDL-C or higher levels of high-density lipoprotein cholesterol (HDL-C) are not always beneficial for the prevention of cardiovascular events. Indeed, it has been shown that subjects with low levels of LDL-C and extremely high levels of HDL-C might have a high risk for cardiovascular events compared with the risk in subjects with normal levels of LDL-C and HDL-C [5,6]. These findings suggest that there is a reverse J-shaped relationship of LDL-C levels with cardiovascular events and a reverse U-shaped relationship of HDL-C levels with cardiovascular events. At present, the role of triglycerides in cardiovascular disease is still uncertain.

Endothelial dysfunction is the initial step for atherosclerosis and contributes to the development and maintenance of atherosclerosis, leading to cardiovascular complications [7,8]. It is thought that dyslipidemia-induced endothelial dysfunction also gradually develops under the condition of poor control of mainly LDL cholesterol levels and finally plays a critical role in the onset of severe cardiovascular events, including myocardial infarction and fatal stroke. It is well known that appropriate interventions such as drug therapy, supplemental therapy and lifestyle modifications, including body weight reduction, aerobic exercise and smoking cessation, restore, improve and augment endothelial function [9,10,11,12,13]. Additionally, in patients with dyslipidemia but not in all patients, treatment with lipid-lowering drugs and lifestyle modifications improves endothelial function [14,15].

Recently, many studies, including our studies, have shown an association of the lipid profile with endothelial function [16,17,18]. We focus on the roles of components of the lipid profile, including LDL-C, HDL-C and triglycerides, in endothelial function.

## 2. Lipid Profile and Endothelial Function

### 2.1. LDL-C and Endothelial Function

A number of epidemiological studies have clearly shown that a high level of LDL-C is a potent predictor of cardiovascular events [1,2,3,4]. It is well known that lipid-lowering therapy, including statins, a combination of statins and ezetimibe, and proprotein convertase subtilisin/kexin type 9 inhibitors, prevents cardiovascular events [19,20,21,22,23].

It is well known that oxidative LDL has significant impacts on the endothelium, the immune system and other components of cardiovascular health [24,25,26,27,28,29]. In the presence of oxidative LDL, oxidative LDL accumulates in the endothelium and the inner lining of blood vessels [7]. This accumulation causes endothelial dysfunction [7]. In addition to impairment of endothelial function, oxidative LDL stimulates the expression of adherence molecules, including intercellular adhesion molecule-1 and vascular cell adhesion molecule-1 on the endothelium, leading to adhesion and migration of immune cells, particularly monocytes, into the arterial wall [25]. This migration may contribute to the formation of atherosclerotic plaques. Oxidative LDL also has direct effects on the immune system [26,27,28]. It activates immune cells such as macrophages and T cells, resulting in the formation of plaques [26,27]. Oxidative LDL also impairs the function of immune cells that are involved in the resolution of inflammation and the repair of damaged tissues [28]. In addition, oxidative LDL inactivates the endothelial nitric oxide synthase (eNOS)/NO pathway [24]. These findings suggest that oxidative LDL contributes to the development and progression of cardiovascular disease through its harmful effects on adherence molecules, eNOS and the immune system in the endothelium. Non-oxidative LDL also plays critical roles in endothelial dysfunction and the development of atherosclerosis, although its effects are not as potent as those of oxidative LDL [29].

What is the optimal level of LDL-C for endothelial function? Are lower LDL-C levels better for endothelial function? Several lines of evidence have shown that dyslipidemia defined by high levels of LDL-C is associated with endothelial dysfunction [16,17]. Even in relatively young men, flow-mediated vasodilation (FMD) as an index of endothelial function was correlated inversely with LDL-C [30]. However, whether there is a significant relationship between LDL-C levels and endothelial function in patients with dyslipidemia is controversial. We also showed that LDL-C level per se was not an independent predictor of endothelial dysfunction assessed by FMD in a general population, including 5314 subjects who were taking lipid-lowering drugs, while LDL-C levels were inversely correlated with FMD in those subjects [18]. In 1346 subjects who were not taking lipid-lowering drugs, LDL-C levels were inversely correlated with FMD, FMD was smaller in subjects with LDL levels of ≥100 mg/dL than in subjects with LDL levels of <100 mg/dL, and FMD values were similar in subjects with LDL levels of 70 to 100 mg/dL and subjects with LDL levels of <70 mg/dL. In 7120 subjects who were not taking lipid-lowering drugs, to evaluate the association of extremely low LDL-C levels with endothelial function, the subjects with LDL-C levels of <70 mg/dL were divided into those with LDL-C levels of <50 mg/dL and those with LDL-C levels of 50–69 mg/dL [31]. FMD values were similar in the subjects with LDL-C levels of <50 mg/dL and subjects with LDL-C levels of 50–69 mg/dL. The relationship between LDL levels and endothelial function, as well as the relationship between LDL levels and cardiovascular events, does not show even lower and even better in limited subjects who were not taking lipid-lowering drugs. Interestingly, there was no significant correlation between LDL levels and FMD in subjects who were taking lipid-lowering drugs. It is thought that statins mask endothelial dysfunction since statins improve endothelial function through their antioxidative and anti-inflammatory effects. Indeed, it is well known that statins have multiple pleiotropic effects and improve endothelial function in patients with dyslipidemia [32,33,34]. We should therefore pay attention to the evaluation of endothelial function under the condition of treatment with lipid-lowering drugs, especially statins. In addition, the roles of LDL-C levels with lipid-lowering therapy and the roles of untreated LDL-C levels in endothelial function should also be separately considered.

Figure 1 shows the relationship of LDL-C levels with FMD using the estimated Lowess smoothed curve in 7120 subjects without lipid-lowering therapy who were enrolled in the Flow-Mediated Dilation Japan (FMD-J) registry and Hiroshima University Vascular Function database [31]. This finding suggests that the optimal level of LDL-C for endothelial function is about 75 mg/dL.

### 2.2. HDL-C and Endothelial Function

It has been established that a low HDL-C level is an independent predictor of cardiovascular events [3,4]. HDL-C per se has various anti-atherosclerotic effects, including activation of endothelial nitric oxide synthase (eNOS), transportation of excess cholesterol from macrophages in the liver and bile, anti-inflammation, anti-oxidation, decrease in oxidative LDL, inhibition of endothelial cell apoptosis, inhibition of platelet aggregation and regulation of adhesion factors [35,36]. In addition, clinical trials showed that an increase in HDL-C levels is related to a decrease in cardiovascular events [37,38]. Two cohort studies, on the other hand, showed that an extremely high HDL-C level was a risk for mortality [39], suggesting that there is a reverse U-shaped relationship between HDL-C levels and mortality.

In general, HDL, also known as “good” cholesterol, has been shown to have beneficial effects on endothelial function [35,36,40,41]. HDL augments and improves endothelial function by activation of the eNOS/NO pathway [36,42]. HDL binds to receptors on the surface of endothelial cells, including scavenger receptor class B type 1 and ATP-binding cassette transporter A1, leading to activation of the PI3K/Akt pathway, which results in the activation of eNOS [36]. HDL inhibits the inactivation of NO by reducing the levels of reactive oxygen species [35,40]. HDL also reduces the expression of adhesion molecules and chemokines that contribute to endothelial dysfunction [35]. HDL augments and improves endothelial function by decreasing inflammation [43]. In addition, HDL removes excess cholesterol from endothelial cells and transports it to the liver for excretion [44]. This prevents the accumulation of cholesterol in the endothelium, which can impair endothelial function.

What is the optimal level of HDL-C for endothelial function? Are higher HDL-C levels better for endothelial function? Many studies have shown that there is a significant positive correlation between HDL-C levels and endothelial function, while this relationship sometimes disappears using multiple regression analyses [30,45,46,47,48], suggesting that HDL-C level is a strong predictor of endothelial function. However, those studies had small numbers of subjects (less than 100 or at most 200) and had limited subjects (e.g., healthy subjects, obese subjects, patients with dyslipidemia, and patients with coronary artery disease) but not a general population.

Kuhn et al. [45] showed that low levels of HDL-C were related to endothelial dysfunction assessed by vascular contractive response to acetylcholine in the coronary artery in 27 patients who underwent coronary angiography. Zeiher et al. [46] also showed that low HDL-C levels were associated with endothelial dysfunction in the coronary artery in 26 patients with angina pectoris. HDL-C levels were correlated with FMD in the brachial artery in 178 healthy military men and 20 young men [30]. Only HDL-C levels were significantly associated with brachial FMD in 63 patients with coronary artery disease and 45 controls [47].

Norimatsu et al. [48] showed that HDL-C was an independent predictor for endothelial dysfunction assessed by the reactive hyperemia index in 191 subjects who were suspected of having coronary artery disease. In our previous study, low HDL-C levels were associated with endothelial dysfunction assessed by FMD in 7682 subjects with and without lipid-lowering therapy, including patients with cardiovascular disease and healthy subjects [49]. Schnell et al. [50] showed that there was an inverse correlation of HDL-C levels with FMD, while there was no significant relationship of HDL-C levels with FMD after adjustment of confounding factors for endothelial function in 40 healthy subjects and 78 patients with dyslipidemia. Interestingly, a rapid increase in HDL-C level with intravenous infusion of reconstituted HDL restored endothelial function assessed by forearm blood flow response to acetylcholine in the brachial artery in patients with dyslipidemia [51]. It is likely that HDL-C level is a predictor for endothelial function.

In 5842 men who were not receiving lipid-lowering therapy, we evaluated endothelial function assessed by FMD in four groups: low HDL-C levels of <40 mg/dL, moderate HDL-C levels of 40 to 59 mg/dL, high HDL-C levels of 60 to 79 md/dL, and extremely high HDL-C levels of ≥80 mg/dL [49]. FMD values were significantly smaller in the low HDL-C group and the extremely high HDL-C group than in the high HDL-C group. There was no significant difference in FMD between the low HDL-C group and the extremely high HDL-C group. Extremely high HDL-C, but not low HDL-C, was significantly associated with a lower quartile of FMD, suggesting that an extremely high level of HDL-C was associated with a reduction in FMD.

In 1719 women who were not receiving lipid-lowering therapy, multiple logistic regression analysis did not reveal a significant association between HDL-C levels and FMD, while there was a positive correlation between HDL-C levels and FMD using a simple regression analysis [52]. When evaluating the relationship between HDL-C levels and FMD, we should therefore pay attention to gender. The relationship between HDL levels and endothelial function, as well as the relationship between HDL levels and cardiovascular events, does not show even higher and even better in subjects who were not receiving lipid-lowering therapy. Future studies are needed to confirm the roles of not only the amount of HDL-C (HDL-C level) but also the function of HDL-C in endothelial function.

Figure 1 shows the relationship of HDL-C levels with FMD using the estimated Lowess smoothed curve in 7120 subjects without lipid-lowering therapy who were enrolled in the FMD-J registry and Hiroshima University Vascular Function database [31]. This finding suggests that the optimal level of HDL-C for endothelial function is about 75 mg/dL.

### 2.3. Triglycerides and Endothelial Function

It is thought that levels of triglycerides are a residual risk for cardiovascular events under the condition of treatment with stains [19,53]. However, it is controversial whether triglycerides per se are an independent risk factor for cardiovascular events [21,54,55,56]. Since statins decrease cholesterol synthesis, they also decrease triglyceride-rich lipoproteins (VLDL), which also contain cholesterol [57,58]. Therefore, these results may indirectly contribute to the reduction of cardiovascular events with statin administration. The Pravastatin or Atorvastatin Evaluation and Infection Therapy-Thrombolysis In Myocardial Infarction 22 trial demonstrated that a decrease in levels of triglycerides of <150 mg/dL reduced the risk of coronary heart disease by 20% compared with the risk with levels of triglycerides of ≥150 mg/dL in patients with acute coronary syndrome in whom LDL-C levels were controlled at <70 mg/dL [21]. Asia-Pacific cohort studies have shown that the level of triglycerides is an independent predictor for cardiovascular events [54]. Iso et al. [55] showed that the level of triglycerides independently predicted the risk of coronary artery disease in a 15.5-year follow-up period after adjustment of levels of total cholesterol and HDL-C in both 4452 Japanese men and 6616 Japanese women. Di Angelantonio et al. [56] reported that there was no significant association of triglycerides with cardiovascular events after adjustment of HDL-C levels. In current guidelines, normal triglyceride levels are defined as levels of less than 150 mg/dL [59,60].

What is the optimal level of triglycerides for endothelial function? Are lower levels of triglycerides better for endothelial function? There are conflicting results concerning the association of triglycerides with endothelial function. Many studies have shown that hypertriglyceridemia impairs endothelial function [18,61,62]. In young healthy men without traditional cardiovascular risk factors, transient hypertriglyceridemia by administration of an emulsion of triglycerides impaired endothelial function assessed by FMD [63]. It has been reported that subjects with hypertriglyceridemia have impaired endothelium-dependent vasodilation in response to acetylcholine in forearm circulation compared to that in healthy subjects [61]. High serum levels of triglycerides are associated with endothelial dysfunction in subjects with metabolic syndrome and patients with coronary artery disease [62,64]. It has also been reported that postprandial elevation of triglyceride levels after a high-fat meal is associated with endothelial dysfunction in healthy subjects [65]. In 4887 subjects who were not receiving lipid-lowering therapy, there was a significant inverse correlation between levels of triglycerides and FMD [18]. After adjustment of conventional cardiovascular risk factors, an increase in the levels of triglycerides was shown to be significantly associated with a decrease in FMD in subjects with levels of triglycerides of <100 mg/dL. In addition, FMD values were highest in subjects with extremely low triglyceride levels of <50 mg/dL. The subjects with extremely low triglyceride levels had fewer cardiovascular risk factors, but FMD remained independently lower even after adjustment for cardiovascular risk factors [66]. It is likely that lower triglyceride levels are better for endothelial function. Unfortunately, it remains unclear whether hypertriglyceridemia is a causal factor for endothelial dysfunction. On the contrary, Chowienczyk et al. [67] and Schnell et al. [50] showed that there is no significant relationship between the levels of triglycerides and endothelial function. There was no significant difference in FMD between patients with hypertriglyceridemia who had no other cardiovascular risk factors and healthy controls. In addition, in hypertriglyceridemia patients with lipoprotein lipase dysfunction who had triglycerides of 1914 ± 1288 mg/dL, endothelial function assessed by forearm vascular response to acetylcholine was not impaired [67].

Figure 1 shows the relationship of triglyceride levels with FMD using the estimated Lowess smoothed curve in 7120 subjects without lipid-lowering therapy who were enrolled in the FMD-J registry and Hiroshima University Vascular Function database [31]. This finding suggests that triglyceride levels are even lower and even better for endothelial function.

### 2.4. Lipid-Lowering Drugs, Endothelial Function and Adverse Effects

It has been shown that lipid-lowering drugs have both anti-inflammatory and antioxidant effects [68,69,70,71,72,73]. Statins are the most common examples of anti-inflammatory drugs [68,69]. Statins not only lower LDL cholesterol by inhibiting cholesterol synthesis but also inhibit the production of cytokines involved in inflammatory responses [68]. Statins also inhibit inflammatory reactions in the vascular wall by improving the function of vascular endothelial cells [69], thereby inhibiting the progression of atherosclerosis. Recent evidence indicates that statins decrease C-reactive protein levels with just 6 weeks of treatment, independent of LDL cholesterol reduction, suggesting that statins possess anti-inflammatory actions [70]. In addition, it was found that statins have antioxidant effects by modulating nuclear factor erythroid-2-related factor 2/heme oxygenase-1 signaling [71]. Fibrates are well known for their antioxidant effects [72,73]. Fibrates reduce circulating levels of lipid peroxides by lowering levels of triglycerides, leading to a decrease in oxidative stress [72]. In addition, fibrates improve systemic antioxidant capacity by stimulating antioxidant synthesis in the liver [73]. Due to these effects, lipid-lowering drugs are expected not only to reduce the risk of atherosclerosis and cardiovascular disease but also to reduce cellular and tissue damage caused by inflammatory responses and oxidative stress.

Lipid-lowering drugs are often safe to use, but side effects have been reported [74,75,76]. The most common side effects of statin drugs include myalgia, muscle damage, liver dysfunction and gastrointestinal symptoms [74]. These side effects depend on the patient’s condition and dosage [74]. Severe side effects are rare, but serious side effects such as muscle syndrome and liver dysfunction can occur [74]. The most common side effects of fibrate drugs include gastrointestinal disorders, liver dysfunction, muscle disorders and cholelithiasis [75]. These side effects also depend on the patient’s condition and dosage [75]. Fibrates are metabolized more slowly in the liver, which may increase the risk of liver damage [75]. Other lipid-lowering medications may increase blood sugar levels and should be used with caution in diabetic patients [76]. They should also not be used or used with caution in pregnant women, lactating women and patients with severe liver disorders. When lipid-lowering drugs are used, proper dosage and administration should be followed, and side effects should be carefully monitored.

## 3. Mechanisms of Endothelial Dysfunction in Dyslipidemia

First of all, we should pay attention to lipid levels before infancy since fetuses born to hypercholesterolemic mothers will have fatty streaks, and the development of atheroma is accelerated in the children of such pregnancies [77,78]. In addition, we should pay attention to lipid profile levels during a person’s infancy. Serum levels of LDL-C are approximately 25 mg/dL, HDL-C levels are approximately 40 mg/dL and levels of triglycerides are approximately 40 mg/dL in infants [79]. Figure 2 shows putative mechanisms of endothelial dysfunction in dyslipidemia.

It has been shown that oxidized LDL or its constituent lipids alter vascular endothelial cell function through suppression of NO production [80], enhancement of the expression of leukocyte adhesion molecules and smooth muscle cell growth factor [81], suppression of endothelial cell migration, inhibition of the production of thrombomodulin [82], and enhancement of the expression of thrombin receptor and tissue factors and induction of apoptosis [83].

There are receptors called scavenger receptors on the endothelial cell membrane [84,85,86,87]. These receptors are responsible for recognizing and removing abnormal intracellular components such as oxidative LDL produced by oxidative stress and glycated proteins [85,86,87]. Representative scavenger receptors include scavenger receptor class A (SR-A) and lectin-like oxidized LDL receptor-1 (LOX-1) [85,86,87]. SR-A is responsible for the recognition of oxidative LDL and its uptake into endothelial cells, and LOX-1 is involved in the recognition of oxidized LDL and glycated proteins [85,86,87]. These scavenger receptors have been reported to be involved in the pathogenesis of atherosclerosis and cardiovascular disease, and inhibition of scavenger receptors may be a potential therapeutic target [84,85,86,87]. In addition, the development of innovative therapies utilizing ligands for scavenger receptors is expected.

There is an interrelationship between Rho-associated kinase activity and the eNOS/NO pathway, and the balance between the two plays an important role in endothelial function [88,89]. It has been shown that Rho-associated kinase directly attenuates the phosphorylation of Akt and eNOS mRNA stability, which leads to the inhibition of eNOS activity, resulting in a decrease in NO production [90]. Retzer et al. [91] reported that oxidized LDL activated the Rho/Rho-associated kinase pathway. In addition, oxidized LDL induced contraction of human umbilical vein endothelial cells through activation of Rho-associated kinase [92]. These findings suggest that an increase in Rho-associated kinase activity and a decrease in NO bioavailability independently and concomitantly contribute to endothelial dysfunction under the condition of a high LDL-C concentration and oxidative LDL-C.

From the aspect of vascular function, low levels of LDL-C and high levels of HDL-C may contribute to the development and maintenance of atherosclerosis, leading to cardiovascular events. However, the precise reasons why low levels of LDL-C and high levels of HDL-C are not better for the prevention of cardiovascular events remain unclear. Subjects with homozygotes for *PCSK9* nonsense mutations had extremely low LDL-C concentrations [93]. Loss-of-function mutation in the angiopoietin-like 3 (*ANGPTL3*) gene is associated with hypolipidemia characterized by low circulating levels of LDL-C, HDL-C and triglycerides [94]. It has been shown that in subjects with homozygotes for *ANGPTL3,* endothelial function assessed by FMD is impaired in spite of low LDL-C levels [95]. Analyses of genetic abnormalities affecting circulating LDL-C levels would enable more specific conclusions concerning the role of low LDL-C levels in endothelial function to be drawn.

It is thought that HDL has beneficial effects on endothelial function through inhibition of LDL oxidation, decrease in oxidative stress, cholesterol efflux from peripheral tissues, prevention of endothelial cell apoptosis, stimulation of endothelial repair processes and activation of eNOS [35,36,40,41]. In addition, HDL decreases inflammatory responses to endothelial cells by protection of LDL from oxidation [42,43]. However, in a clinical setting, the effects of HDL on endothelial function are not always constant [30,45,46,47,48,49,52]. Interestingly, HDL isolated from patients with type 2 diabetes or patients with cardiovascular disease had abolished the ability to stimulate eNOS activity and its ability to prevent NF-κB activation stimulated by tumor necrosis factor alpha was reduced in endothelial cells, while HDL isolated from healthy subjects activated the eNOS/NO pathway and reduced oxidative stress stimuli in endothelial cells [96]. Genome-wide association studies have revealed that novel loci are associated with HDL-C levels [97]. Loss-of-function mutations in genes, including a lipase gene (*LIPG*), scavenger receptor class B number 1 (*SCARB1*) and cholesteryl ester transfer protein (*CETP*), induce the condition of extremely high HDL-C levels [98,99,100,101]. The *LIPG* gene encodes endothelial lipase that mediates HDL catabolism [99]. The *SRARB1* gene encodes scavenger receptor class B type 1, which is an HDL receptor, and promotes the uptake of HDL cholesteryl esters into hepatocytes [98]. *CETP* deficiency plays a critical role in the increase in HDL-C levels in Japan [100,101]. Yamashita et al. [102]. showed that subjects with CETP deficiency had an accumulation of apolipoprotein E-rich HDLs. It has been shown that HDL-C isolated from subjects who are carriers of *CETP* deficiency decreased eNOS activity [103]. It is likely that dysfunctional HDL inactivates the eNOS/NO pathway through inhibition of CETP-induced increases in oxidative stress and inflammation. When considering the effects of HDL on endothelial function, attention should be given to HDL function and circulating HDL-C levels.

It is well known that there are gender differences in HDL-C levels. HDL-C level was associated with endothelial function in men but not in women [49,52]. Possible reasons for the gender-opposite findings regarding the association of HDL-C levels with endothelial function are differences in *CETP* activity and levels of estrogen and androgen. Indeed, some single nucleotide polymorphisms in *CETP* are related to different circulating HDL levels in men and women [101,104]. It has been shown that HDL from women enhances eNOS activity, while HDL from men has little effect on eNOS activity [105]. Confounding factors for endothelial function such as smoking, alcohol consumption and exercise other than genetic variation and differences in levels of estrogen and androgen also may contribute to the different associations of HDL-C with endothelial function in men and women.

It is unclear whether there is a direct causal relationship between triglycerides and endothelial dysfunction. The concentration of circulating triglycerides corresponds to the concentration of triglyceride-rich lipoproteins (TRLs) and their residues [106]. Lipoprotein lipase (LPL) hydrolyzes TRLs to produce TRL remnants, leading to apoptosis in endothelial cells through the production of cytokines, including tumor necrosis factor-α and interleukin-1β [107,108]. In addition, TRL remnants promote endothelial dysfunction by an increase in endothelial cell inflammatory response to stimuli [109]. In a clinical setting, endothelial function was preserved in patients with severe hypertriglyceridemia associated with LPL dysfunction [67]. These findings suggest that LPL-induced hydrolysis of TRLs contributes to endothelial dysfunction. In high triglyceride states, the transfer of triglycerides from TRLs to HDL particles by cholesteryl ester transfer protein is enhanced [110]. Enrichment of HDL with triglycerides enhanced HDL-associated apolipoprotein A-I clearance, resulting in a decrease in circulating HDL-C concentrations [110]. In addition, it has been shown that an increase in triglycerides is related to a decrease in LDL size and an increase in the number of LDL particles [111]. Under the condition of hypertriglyceridemia, it is likely that small dense LDL particles cause endothelial dysfunction through susceptibility to oxidation of small dense LDL particles and an increase in cell adhesion molecules [112,113].

Endothelial progenitor cells (EPCs) may play an important role in dyslipidemia [114,115,116]. EPCs are differentiated from bone-marrow-derived stem or progenitor cells and have the ability to differentiate into vascular wall cells such as vascular endothelial cells and vascular smooth muscle cells [117,118]. EPCs have the ability to repair vascular endothelial cell damage and injury caused by dyslipidemia [114,116]. Continued dysfunction of vascular endothelial cells can lead to stiffening of the vessel wall and cause atherosclerosis [117,118]. It is thought that EPCs prevent the progression of atherosclerosis by regenerating and repairing endothelial cells in such situations. EPCs also have anti-inflammatory and antioxidant actions against oxidative stress and inflammation caused by dyslipidemia [114,115,116]. Through these functions, EPCs play an important role in the maintenance of vascular health. Because of their ability to differentiate into vascular endothelial cells and smooth muscle cells, EPCs are being investigated for the treatment of atherosclerosis and cardiovascular disease [117,118]. Many currently available drugs, including lipid-lowering drugs that impact cardiovascular morbidity and mortality, have shown a positive effect on EPC biology [119].

## 4. Conclusions

There is a reverse J-shaped relationship of LDL-C level with FMD, a reverse U-shaped relationship of HDL-C level with FMD and a linear inverse correlation of triglyceride level with FMD in individuals who are not receiving lipid-lowering therapy. Future studies are needed to evaluate in detail the association of each compartment of the lipid profile with endothelial function. Assessment of endothelial function in dyslipidemia is one of the most important issues for the prevention and treatment of cardiovascular diseases. Future research directions may include the following. First, we should pay attention to the interaction between endothelial cells and lipid metabolism. Endothelial cells express enzymes and receptors involved in lipid metabolism, and the effects of dyslipidemia on endothelial function are complex. In future research, the mechanisms of endothelial dysfunction caused by dyslipidemia should be investigated through a detailed analysis of the interactions between endothelial cells and lipid metabolism. Second, various drugs such as statins and fibrates are used to treat dyslipidemia, but these drugs also have effects on endothelial function. In the future, the development of therapeutic agents specifically for endothelial dysfunction caused by dyslipidemia is required. Third, measurements of blood flow responses to vasoactive agents using strain-gauge plethysmography, FMD and reactive hyperemia index are currently used to evaluate endothelial function. Unfortunately, it is unclear whether those methods are adequate for the assessment of endothelial function in patients with dyslipidemia. The establishment of a specific method for the measurement of endothelial function in patients with dyslipidemia would enable more specific conclusions concerning the roles of lipid metabolism in endothelial function to be drawn.

## Figures and Tables

**Figure 1 cells-12-01293-f001:**
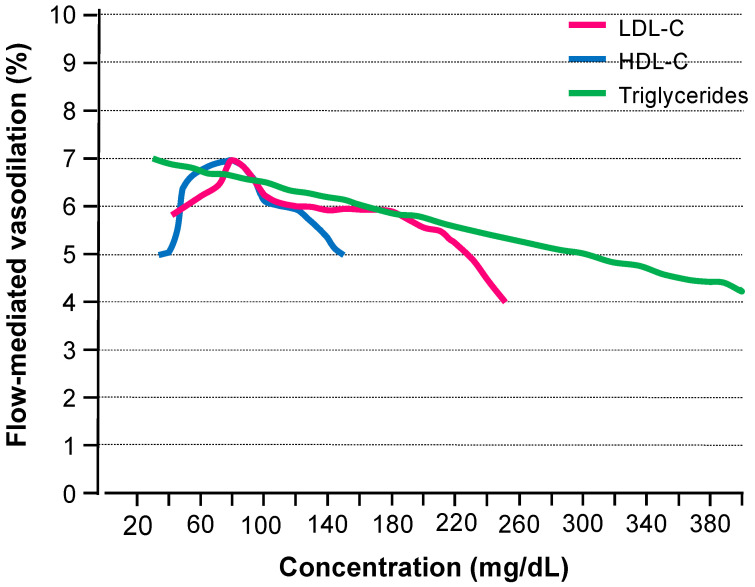
Line graphs show the relationships of low-density lipoprotein cholesterol (LDL-C), high-density lipoprotein cholesterol (HDL-C) and triglycerides with flow-mediated vasodilation (FMD) using the estimated Lowess smoothed curve in 10,073 subjects without lipid-lowering therapy who were enrolled in the FMD-J registry and Hiroshima University database. (Modified and quoted from Reference [31]).

**Figure 2 cells-12-01293-f002:**
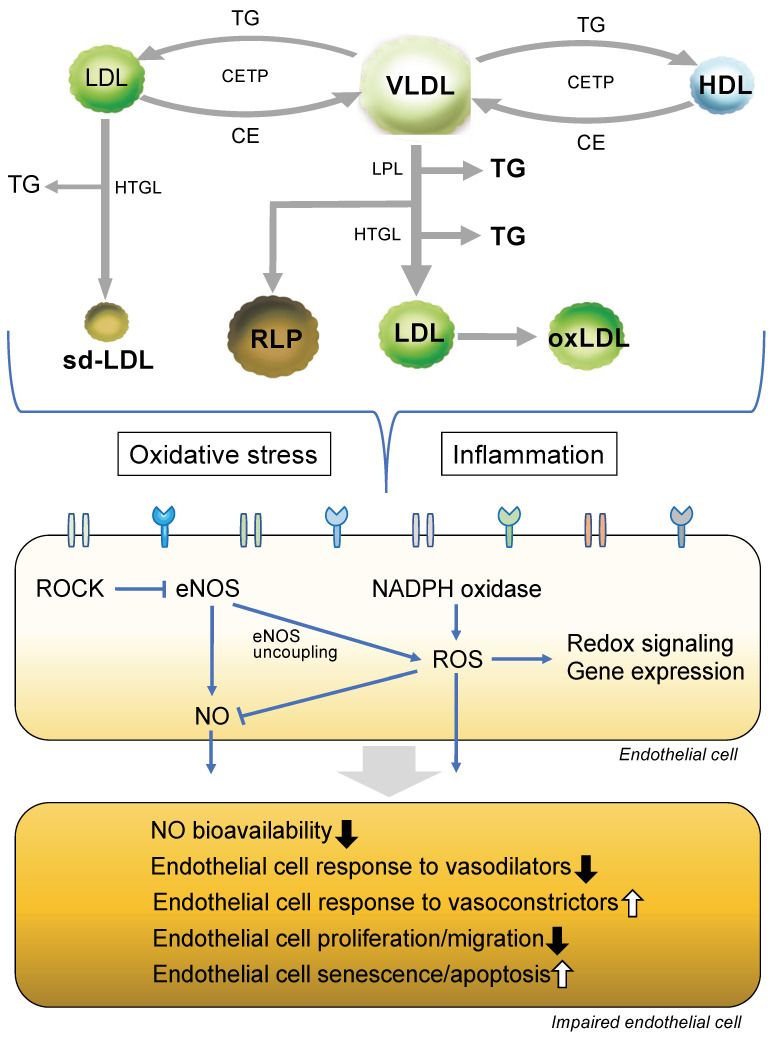
Putative mechanisms of dyslipidemia-induced endothelial dysfunction. LDL indicates low-density lipoprotein; TG, triglycerides; VLDL, very LDL; CETP, cholesteryl ester transfer protein; HDL, high-density lipoprotein; HTGL, hepatic triacylglycerol lipase; LPL, lipoprotein lipase; sd-LDL, small dense LDL; RLP, remnant-like particles; oxLDL, oxidative LDL; ROCK, Rho-associated kinase; eNOS, endothelial nitric oxide synthase; NADPH, nicotinamide adenine dinucleotide phosphate; ROS, reactive oxygen species; NO, nitric oxide.

## Data Availability

The raw data supporting the conclusions of this article will be made available by the authors, without undue reservation.

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
