# Peer review of "Endothelial Function in Dyslipidemia: Roles of LDL-Cholesterol, HDL-Cholesterol and Triglycerides"

_cells, 2023, doi:10.3390/cells12091293_

Round 1

Reviewer 1 Report

In this review article, the author reviewed the roles of LDL-cholesterol, HDL-cholesterol and triglycerides in endothelial function in dyslipidemia. Since dyslipidemia is associated with endothelial dysfunction and endothelial dysfunction is the initial step for atherosclerosis, resulting in cardiovascular complications, it is important to break the process of endothelial dysfunction into cardiovascular complications in patients with dyslipidemia. Lipid-lowering therapy enables the improvement of endothelial function in patients with dyslipidemia. The relationships of components of a lipid profile such as low-density lipoprotein cholesterol, high-density lipoprotein cholesterol and triglycerides with endothelial function are not simple. In this review, the author focuses on the roles of component of a lipid profile in endothelial function. However, future studies are needed to evaluate in detail the association of each comportment of the lipid profile with endothelial function. It is an interesting review article. Specific comments:

1.          Please break up the long paragraph (Page 4) into smaller sections.  

2.          The iconographies and tables are particularly welcome for the review article to attract the readers. However, Figure 1 is modified and quoted from reference #25, the author should obtain permission to reproduce any published material (figures, schemes, tables, or any extract of a text) that does not fall into the public domain, or for which they do not hold the copyright. Permission should be requested by the authors from the copyright holder. Please refer to https://www.mdpi.com/authors/rights for more details. If you have obtained permission, please upload related files in the below frame; if the manuscript does not involve copyright issues, please choose “no copyright issues”.

3.          There is a reverse J-shaped relationship between LDL-C level with FMD, and a reverse U-shaped relationship between HDL-C level with FMD. Why and how lower levels of LDL-C and high levels of HDL-C are not better for the prevention of cardiovascular events?

4.          It is well known that statins have multiple pleiotropic effects and improve endothelial function in patients with dyslipidemia. The vascular inflammatory response is a complex process that leads to thrombus formation, angiogenesis, neointimal thickening, and atherosclerosis. Recent evidence indicates that statins decrease C-reactive protein levels in just 6 weeks of treatment, independent of LDL cholesterol reduction, and suggests that statins possess anti-inflammatory actions. In addition, the antioxidant effects of statins were found by modulating Nrf2/HO-1 signaling. Please briefly discuss the anti-inflammatory and antioxidant actions of the lipid-lowing drugs.

5.          The receptors (such as scavenger receptors) on endothelial cell member should be specified and indicated.

6.          References #30 and #73 are duplicates.

7.          Are there any potential side effects of the lipid-lowing approach? Please discuss the safety of lipid-lowing drugs.

8.          The discovery of endothelial progenitor cells (EPCs), with their ability to replace old and injured cells and differentiate into healthy and functional mature endothelial cells, has shifted our view of atherosclerosis as an incurable disease. Many currently available drugs (including lipid-lowing drugs) that impact cardiovascular morbidity and mortality have shown a positive effect on EPC biology. Please also briefly discuss the EPCs in dyslipidemia.

9.          The authors should discuss the severe limitations of their approach and give future directions for research in the field of endothelial function in dyslipidemia.

Author Response

Manuscript ID: cells-2326404 R1                               Reviewer 1

I would like to thank the reviewer for the helpful comments and hope that we have now produced a more balanced and better account of our work.

Specific comments:

  1. Please break up the long paragraph (Page 4) into smaller sections.

Response: In accordance with the reviewer’s suggestion, the long paragraph (page 4) has been broken up into smaller sections.

  1. The iconographies and tables are particularly welcome for the review article to attract the readers. However, Figure 1 is modified and quoted from reference #25, the author should obtain permission to reproduce any published material (figures, schemes, tables, or any extract of a text) that does not fall into the public domain, or for which they do not hold the copyright. Permission should be requested by the authors from the copyright holder. Please refer to https://www.mdpi.com/authors/rights for more details. If you have obtained permission, please upload related files in the below frame; if the manuscript does not involve copyright issues, please choose “no copyright issues”.

Response: Since the manuscript does not involve copyright issues, I chose “no copyright issues”.

  1. There is a reverse J-shaped relationship between LDL-C level with FMD, and a reverse U-shaped relationship between HDL-C level with FMD. Why and how lower levels of LDL-C and high levels of HDL-C are not better for the prevention of cardiovascular events?

Response: From the aspect of vascular function, low levels of LDL-C and high levels of HDL-C may contribute to the development and maintenance of atherosclerosis, leading to cardiovascular events. However, the precise reasons why low levels of LDL-C and high levels of HDL-C are not better for the prevention of cardiovascular events remain unclear. These comments have been incorporated into Section 3 (lines 321-324).

  1. It is well known that statins have multiple pleiotropic effects and improve endothelial function in patients with dyslipidemia. The vascular inflammatory response is a complex process that leads to thrombus formation, angiogenesis, neointimal thickening, and atherosclerosis. Recent evidence indicates that statins decrease C-reactive protein levels in just 6 weeks of treatment, independent of LDL cholesterol reduction, and suggests that statins possess anti-inflammatory actions. In addition, the antioxidant effects of statins were found by modulating Nrf2/HO-1 signaling. Please briefly discuss the anti-inflammatory and antioxidant actions of the lipid-lowing drugs.

Response: It has been shown that lipid-lowering drugs have both anti-inflammatory and antioxidant effects [new refs 71-76]. Statins are the most common examples of anti-inflammatory drugs [new refs 71 and 72]. Statins not only lower LDL cholesterol by inhibiting cholesterol synthesis but also inhibit the production of cytokines involved in inflammatory responses [new ref 71]. Statins also inhibit inflammatory reactions in the vascular wall by improving the function of vascular endothelial cells, thereby inhibiting the progression of atherosclerosis [new ref 72]. Recent evidence indicates that statins decrease C-reactive protein levels with just 6 weeks of treatment, independent of LDL cholesterol reduction, suggesting that statins possess anti-inflammatory actions [new ref 73]. In addition, it was found that statins have antioxidant effects by modulating Nrf2/HO-1 signaling [new ref 74]. Fibrates are well known for their antioxidant effects [new refs 75 and 76]. Fibrates reduce circulating levels of lipid peroxides by lowering levels of triglycerides, leading to a decrease in oxidative stress [new ref 75]. In addition, fibrates improve systemic antioxidant capacity by stimulating antioxidant synthesis in the liver [new ref 76]. Due to these effects, lipid-lowering drugs are expected not only to reduce the risk of atherosclerosis and cardiovascular disease but also to reduce cellular and tissue damage caused by inflammatory responses and oxidative stress. These comments have been incorporated into Section 2.4.

  1. The receptors (such as scavenger receptors) on endothelial cell member should be specified and indicated.

Response: There are receptors called scavenger receptors on the endothelial cell membrane [new refs 87-90]. These receptors are responsible for recognizing and removing abnormal intracellular components such as oxidative LDL produced by oxidative stress and glycated proteins [new refs 88-90]. Representative scavenger receptors include scavenger receptor class A (SR-A) and lectin-like oxidized LDL receptor-1 (LOX-1) [new refs 88-90]. SR-A is responsible for the recognition of oxidative LDL and its uptake into endothelial cells and LOX-1 is involved in the recognition of oxidized LDL as well as glycated proteins [new refs 88-90]. These scavenger receptors have been reported to be involved in the pathogenesis of atherosclerosis and cardiovascular disease, and inhibition of scavenger receptors may be a potential therapeutic target [new refs 87-90]. In addition, the development of innovative therapies utilizing ligands for scavenger receptors is expected. These comments have been incorporated into Section 3 (lines 299-309).

  1. References #30 and #73 are duplicates.

Response: Reference 73 in the previous version has been deleted.

  1. Are there any potential side effects of the lipid-lowing approach? Please discuss the safety of lipid-lowing drugs.

Response: Lipid-lowering drugs are often safe to use, but side effects have been reported [new refs 77-79]. The most common side effects of statin drugs include myalgia, muscle damage, liver dysfunction, and gastrointestinal symptoms [new ref 77]. These side effects depend on the patient's condition and dosage. Severe side effects are rare, but serious side effects such as muscle syndrome and liver dysfunction can occur [new ref 77]. The most common side effects of fibrate drugs include gastrointestinal disorders, liver dysfunction, muscle disorders, and cholelithiasis [new ref 77]. These side effects also depend on the patient's condition and dosage. Fibrates are metabolized more slowly in the liver, which may increase the risk of liver damage [new ref 78]. Other lipid-lowering medications may increase blood sugar levels and should be used with caution in diabetic patients [new ref 79]. They should also not be used or used with caution in pregnant women, lactating women, and patients with severe liver disorders. When lipid-lowering drugs are used, proper dosage and administration should be followed and side effects should be carefully monitored. These comments have been incorporated into Section 2.4.

  1. The discovery of endothelial progenitor cells (EPCs), with their ability to replace old and injured cells and differentiate into healthy and functional mature endothelial cells, has shifted our view of atherosclerosis as an incurable disease. Many currently available drugs (including lipid-lowing drugs) that impact cardiovascular morbidity and mortality have shown a positive effect on EPC biology. Please also briefly discuss the EPCs in dyslipidemia.

Response: Endothelial progenitor cells (EPCs) may play an important role in dyslipidemia [new refs 118-120]. EPCs are differentiated from bone marrow-derived stem or progenitor cells and have the ability to differentiate into vascular wall cells such as vascular endothelial cells and vascular smooth muscle cells [new refs 121 and 122]. EPCs have the ability to repair vascular endothelial cell damage and injury caused by dyslipidemia [new refs 118-120]. Continued dysfunction of vascular endothelial cells can lead to stiffening of the vessel wall and cause atherosclerosis [new refs 121 and 122]. It is thought that EPCs prevent the progression of atherosclerosis by regenerating and repairing endothelial cells in such situations. EPCs also have anti-inflammatory and antioxidant actions against oxidative stress and inflammation caused by dyslipidemia [new refs 118-120]. Through these functions, EPCs play an important role in the maintenance of vascular health. Because of their ability to differentiate into vascular endothelial cells and smooth muscle cells, EPCs are being investigated for treatment of atherosclerosis and cardiovascular disease [new refs 121 and 122]. Many currently available drugs including lipid-lowering drugs that impact cardiovascular morbidity and mortality have shown a positive effect on EPC biology [new ref 123]. These comments have been incorporated into Section 3 (lines 387-401).

  1. The authors should discuss the severe limitations of their approach and give future directions for research in the field of endothelial function in dyslipidemia.

Response: Assessment of endothelial function in dyslipidemia is one of the most important issues for the prevention and treatment of cardiovascular diseases. Future research directions may include the following. First, we should pay attention to the interaction between endothelial cells and lipid metabolism. Endothelial cells express enzymes and receptors involved in lipid metabolism, and the effects of dyslipidemia on endothelial function are complex. In future research, the mechanisms of endothelial dysfunction caused by dyslipidemia should be investigated through detailed analysis of the interactions between endothelial cells and lipid metabolism. Second, various drugs such as statins and fibrates are used to treat dyslipidemia, but these drugs also have effects on endothelial function. In the future, the development of therapeutic agents specifically for endothelial dysfunction caused by dyslipidemia is required. Third, measurements of blood flow responses to vasoactive agents using strain-guage plethysmography, FMD and reactive hyperemia index are currently used to evaluate endothelial function. Unfortunately, it is unclear whether those methods are adequate for assessment of endothelial function in patients with dyslipidemia. The establishment of a specific method for measurement of endothelial function in patients with dyslipidemia would enable more specific conclusions concerning the roles of lipid metabolism in endothelial function to be drawn. These comments have been incorporated into the Conclusions section (lines 407-423).

Reviewer 2 Report

This review is about the contribution of HDL, LDL cholesterols, and Triglycerides to the development of endothelial dysfunction.

However, the authors do not explain in detail the role of these lipids in this alteration, they gathered information about these lipids, talked about lipid-lowering or increasing lipid drugs, and linked this information to CVD, but lipids were not linked to the pathophysiology of the endothelial function. The authors reviewed the effect of mentioned lipids on clinical approaches (FMD); clinical parameter of endothelial dysfunction.

In general, the manuscript has a lot of information that is not related to the main topic. Thus, maybe a title change should be done.

Details

Line 31, references 5, 6 are not related to high levels of HDL, please correct or add the correct reference

Line 33: “It remains unclear whether lower levels of triglycerides are better for prevention of cardiovascular events.  Nothing is previously mentioned about triglycerides (TG), therefore this phrase is out of context. Actually, nowadays the role of TG in cardiovascular disease is still uncertain.

Line 38: “endothelial dysfunction also gradually develops under the condition of poor control of cholesterol levels”. Concerning this affirmation not all cholesterol is related to this, mainly LDL cholesterol.

Line 41, reference is needed.

Section 2. Line52, references 1-4 are not in the same format as the others

Line 54, if the goal of this phrase is to update the existing information, new technologies as inclisiran and PCSK9 antibodies, are used to diminish LDL concentration. However, the focus of this section is showing the effect of LDL on endothelial function. Thus, this information is not relevant.

 Line 63: “However, a recent epidemiological study has shown that LDL-C levels of <70 mg/dL increase the incidence of cardiovascular events in subjects who are not receiving lipid-lowering therapy”. This article does not show the LDL values and the incidence in CVD, they show the levels of LDL and the risk of intracerebral hemorrhage. Concerning this, the origin of the intracerebral hemorrhage the linked to CVD, but in the article it is not shown. Therefore, these results should be interpreted carefully.

Section 2.1 The focus of this section is interesting but is mainly related to LDL concentration and Flow- mediated vasodilation as a measure of endothelium function, and how lipid-lowering drugs can impact the last parameter. However, as the authors mentioned earlier, endothelium function is the start of CVD, and thus, not only the bad functioning of the endothelium should be reviewed. It is necessary also complement and review how oxidized LDL can impact the endothelium, the effect of these lipoproteins on adherence molecules, eNOS, immune components, etc.  In addition, the same information should be included related to the LDL non oxidized.

Line 119. “Two cohort studies, on the other hand, showed that an extremely high HDL-C level was harmful for cardiovascular events and mortality [33]. This study does not point out this association, it is done with extremely high levels of HDL and all-cause mortality. Please correct this information.

Line 120 “These findings suggest that there is a reverse U- shaped relationship between HDL-C levels and cardiovascular events same comment as before, please correct the information and use a reference for this phrase.

Lines 121 to 127 are related to a drug that increases HDL cholesterol, a topic that deviates from the main theme of the review.

As section 2.1, 2.2 should be complemented with other information related to endothelial function as mentioned above.

As it is mentioned in section 2.3, the participation of TG in CVD development is controversial because is not completely clear if this lipid has a direct impact on CVD. To point out this, the article included here quote: “The Pravastatin or Atorvastatin Evaluation and Infection Therapy-Thrombolysis In Myocardial Infarction 22 trial demonstrated that a decrease in levels of triglycerides of <150 mg/dL reduced the risk of coronary heart disease by 20% compared with the risk with levels of triglycerides of ≥150 mg/dL in patients with acute coronary syndrome in whom LDL-C levels were controlled at <70 mg/dL [21].” Should be discussed carefully regarding TG levels. Since statins decrease cholesterol synthesis, also lipoproteins that contain more TG, which also contains cholesterol (VLDL) will diminish. Therefore, these results indirectly could contribute to CVD.

The above idea it is mentioned in line 302: “It is unclear whether there is the direct causal relationship between triglycerides and 302 endothelial dysfunction”. This should be mentioned at the beginning of section 2.3.

Section 3. Nowadays we know that we should pay attention to lipids levels before infancy, this should be followed during pregnancy because fetuses from hipercholesterolemic mothers are born with fatty streaks (Napolli 1997, 1999), and in children from these kinds of pregnancies show accelerated development of atheromas. Please modify this phrase. However, this paragraph is out of the context of the main theme.

Lines 242 to 258 this information should be included when LDL are mentioned.

Author Response

Manuscript ID: cells-2326404 R1                             Reviewer 2

I would like to thank the reviewer for the helpful comments and hope that we have now produced a more balanced and better account of our work.

Details

Line 31, references 5, 6 are not related to high levels of HDL, please correct or add the correct reference.

Response: References 5 and 6 have been changed to appropriate new references 5 and 6 (line 31).

Line 33: “It remains unclear whether lower levels of triglycerides are better for prevention of cardiovascular events”. Nothing is previously mentioned about triglycerides (TG), therefore this phrase is out of context. Actually, nowadays the role of TG in cardiovascular disease is still uncertain.

Response: In accordance with reviewer’s appropriate suggestion, the sentence “It remains unclear whether lower levels of triglycerides are better for prevention of cardiovascular events.” has been changed to “At present, the role of triglycerides in cardiovascular disease is still uncertain.” (line 33).

Line 38: “endothelial dysfunction also gradually develops under the condition of poor control of cholesterol levels”. Concerning this affirmation not all cholesterol is related to this, mainly LDL cholesterol.

Response: In accordance with reviewer’s appropriate suggestion, the sentence “endothelial dysfunction also gradually develops under the condition of poor control of cholesterol levels.” has been changed to “endothelial dysfunction also gradually develops under the condition of poor control of mainly LDL cholesterol levels.” (line 38).

Line 41, reference is needed.

Response: Since the sentence “It is thought that dyslipidemia-induced endothelial dysfunction also gradually develops under the condition of poor control of mainly LDL cholesterol levels and finally plays a critical role in the onset of severe cardiovascular events including myocardial infarction and fatal stroke.” starts with " It is thought that ,,,." a reference was not cited.

Section 2. Line 52, references 1-4 are not in the same format as the others.

Response: References 1-4 have been changed to the same format.

Line 54, if the goal of this phrase is to update the existing information, new technologies as inclisiran and PCSK9 antibodies, are used to diminish LDL concentration. However, the focus of this section is showing the effect of LDL on endothelial function. Thus, this information is not relevant.

Response: In accordance with reviewer’s appropriate suggestion, information on new technologies as inclisiran and PCSK9 antibodies that are used to diminish LDL concentration has been deleted in this section.

Line 63: “However, a recent epidemiological study has shown that LDL-C levels of <70 mg/dL increase the incidence of cardiovascular events in subjects who are not receiving lipid-lowering therapy”. This article does not show the LDL values and the incidence in CVD, they show the levels of LDL and the risk of intracerebral hemorrhage. Concerning this, the origin of the intracerebral hemorrhage the linked to CVD, but in the article it is not shown. Therefore, these results should be interpreted carefully.

Response: I agree with the reviewer’s comment. The sentence “However, a recent epidemiological study has shown that LDL-C levels of <70 mg/dL increase the incidence of cardiovascular events in subjects who are not receiving lipid-lowering therapy.” has been deleted in this section.

Section 2.1 The focus of this section is interesting but is mainly related to LDL concentration and Flow- mediated vasodilation as a measure of endothelium function, and how lipid-lowering drugs can impact the last parameter. However, as the authors mentioned earlier, endothelium function is the start of CVD, and thus, not only the bad functioning of the endothelium should be reviewed. It is necessary also complement and review how oxidized LDL can impact the endothelium, the effect of these lipoproteins on adherence molecules, eNOS, immune components, etc. In addition, the same information should be included related to the LDL non oxidized.

Response: In accordance with reviewer’s appropriate suggestion, the following sentences have been incorporated into Section 2.1 (lines 55-72): “It is well known that oxidative LDL has significant impacts on the endothelium, the immune system, and other components of cardiovascular health [new refs 24-29]. In the presence of oxidative LDL, oxidativeLDL accumulates in the endothelium and the inner lining of blood vessels [ref 7]. This accumulation causes endothelial dysfunction [ref 7]. In addition to impairment of endothelial function, oxidative LDL stimulates the expression of adherence molecules including intercellular adhesion molecule-1 (ICAM-1) and vascular cell adhesion molecule-1 (VCAM-1) on the endothelium, leading to adhesion and migration of immune cells, particularly monocytes, into the arterial wall [new ref 25]. This migration may contribute to the formation of atherosclerotic plaques [new refs 26-28]. Oxidative LDL also has direct effects on the immune system [new refs 26-28]. It activates immune cells such as macrophages and T cells, resulting in the formation of plaques [new refs 26 and 27]. Oxidative LDL also impairs the function of immune cells that are involved in the resolution of inflammation and the repair of damaged tissues [new ref 28]. In addition, oxidative LDL inactivates the endothelial nitric oxide synthase (eNOS)/NO pathway [new ref 24]. These findings suggest that oxidative LDL contributes to the development and progression of cardiovascular disease through its harmful effects on adherence molecules, eNOS, and the immune system in the endothelium. Non-oxidative LDL also plays critical roles in endothelial dysfunction and the development of atherosclerosis, although its effects are not as potent as those of oxidative LDL [new ref 29].”

Line 119. “Two cohort studies, on the other hand, showed that an extremely high HDL-C level was harmful for cardiovascular events and mortality [33]”. This study does not point out this association, it is done with extremely high levels of HDL and all-cause mortality. Please correct this information.

Line 120 “These findings suggest that there is a reverse U- shaped relationship between HDL-C levels and cardiovascular events” same comment as before, please correct the information and use a reference for this phrase.

Response: In accordance with reviewer’s appropriate suggestion, the sentences “Two cohort studies, on the other hand, showed that an extremely high HDL-C level was harmful for cardiovascular events and mortality. These findings suggest that there is a reverse U-shaped relationship between HDL-C levels and cardiovascular events.” have been corrected to “Two cohort studies, on the other hand, showed that an extremely high HDL-C level was a risk for mortality, suggesting that there is a reverse U-shaped relationship between HDL-C levels and mortality.”(lines 123-126).

Lines 121 to 127 are related to a drug that increases HDL cholesterol, a topic that deviates from the main theme of the review.

Response: Information on a drug that increases HDL cholesterol has been deleted (previous lines 121 to 127).

As section 2.1, 2.2 should be complemented with other information related to endothelial function as mentioned above.

Response: In accordance with reviewer’s appropriate suggestion, the following sentences have been incorporated into the Section 2.2 (lines 127-138: “In general, HDL, also known as "good" cholesterol, has been shown to have beneficial effects on endothelial function [new refs 36, and 40-42]. HDL augments and improves endothelial function by activation of the eNOS/NO pathway [new refs 36 and 43]. HDL binds to receptors on the surface of endothelial cells, including scavenger receptor class B type 1 and ATP-binding cassette transporter A1, leading to activation of PI3K/Akt pathway, which results in activation of eNOS [new ref 36]. HDL inhibits the inactivation of NO by reducing the levels of reactive oxygen species (ROS) [new refs 40 and 41]. HDL also reduces the expression of adhesion molecules and chemokines that contribute to endothelial dysfunction [new ref 41]. HDL augments and improves endothelial function by decreasing in inflammation [new ref 44]. In addition, HDL removes excess cholesterol from endothelial cells and transports it to the liver for excretion [new ref 45]. This prevents the accumulation of cholesterol in the endothelium, which can impair endothelial function.”).

As it is mentioned in section 2.3, the participation of TG in CVD development is controversial because is not completely clear if this lipid has a direct impact on CVD. To point out this, the article included here quote: “The Pravastatin or Atorvastatin Evaluation and Infection Therapy-Thrombolysis In Myocardial Infarction 22 trial demonstrated that a decrease in levels of triglycerides of <150 mg/dL reduced the risk of coronary heart disease by 20% compared with the risk with levels of triglycerides of ≥150 mg/dL in patients with acute coronary syndrome in whom LDL-C levels were controlled at <70 mg/dL [21].” Should be discussed carefully regarding TG levels. Since statins decrease cholesterol synthesis, also lipoproteins that contain more TG, which also contains cholesterol (VLDL) will diminish. Therefore, these results indirectly could contribute to CVD. The above idea it is mentioned in line 302: “It is unclear whether there is the direct causal relationship between triglycerides and endothelial dysfunction”. This should be mentioned at the beginning of section 2.3.

Response: I agree with the reviewer’s comment. The following sentences have been incorporated into Section 2.3 (lines 194-196): “Since statins decrease cholesterol synthesis they also decrease triglyceride-rich lipoproteins (VLDL), which also contain cholesterol [new refs 59 and 60]. Therefore, these results may indirectly contribute to the reduction of cardiovascular events with statin administration.”.

Section 3. Nowadays we know that we should pay attention to lipids levels before infancy, this should be followed during pregnancy because fetuses from hipercholesterolemic mothers are born with fatty streaks (Napolli 1997, 1999), and in children from these kinds of pregnancies show accelerated development of atheromas. Please modify this phrase. However, this paragraph is out of the context of the main theme. Lines 242 to 258 this information should be included when LDL are mentioned.

Response: In accordance with reviewer’s appropriate suggestion, the following sentence has been incorporated into Section 3 (lines 274-276): “In addition, we should pay attention to lipid levels before infancy since fetuses born to hypercholesterolemic mothers will have fatty streaks and the development of atheroma is accelerated in the children of such pregnancies (Napoli et al., J Clin Invest. 1997; 100: 2680-2690 and Napoli et al., Lancet. 1999; 354: 1234-1241 as new refs 80 and 81).”.

Round 2

Reviewer 1 Report

In this review article, the author reviewed the roles of LDL-cholesterol, HDL-cholesterol and triglycerides in endothelial function in dyslipidemia. Since dyslipidemia is associated with endothelial dysfunction and endothelial dysfunction is the initial step for atherosclerosis, resulting in cardiovascular complications, it is important to break the process of endothelial dysfunction to cardiovascular complications in patients with dyslipidemia. Lipid-lowering therapy enables improvement of endothelial function in patients with dyslipidemia. The relationships of component of a lipid profile such as low-density lipoprotein cholesterol, high-density lipoprotein cholesterol and triglycerides with endothelial function are not simple. In this review, the author focus on the roles of component of a lipid profile in endothelial function. However, future studies are needed to evaluate in detail the association of each comportment of the lipid profile with endothelial function. It is an interesting review article. The revision of the manuscript is much improved, no additional comments.

Reviewer 2 Report

After the changes done by the authors I do not have further observations